# Lifestyle, sick leave and work ability among Norwegian employees with asthma—A population-based cross-sectional survey conducted in Telemark County, Norway

**Marit Müller De Bortoli**[1,2☯¤]*, **Anne Kristin Møller Fell**[2☯], **Martin Veel Svendsen**[2☯], **Paul K. Henneberger**[3‡], **Johny Kongerud**[4,5‡], **Inger M. Oellingrath**[1☯]

1 Department of Nursing and Health Sciences, Faculty of Health and Social Sciences, University of South-Eastern, Porsgrunn, Vestfold and Telemark, Norway, 2 Department of Occupational and Environmental Medicine, Telemark Hospital, Skien, Vestfold and Telemark, Norway, 3 National Institute for Occupational Safety and Health, Centers for Disease Control and Prevention, Morgantown, West Virginia, United States of America, 4 Institute of Clinical Medicine, Faculty of Medicine, University of Oslo, Oslo, Norway, 5 Department of Respiratory Medicine, Oslo University Hospital, Oslo, Norway

☯ These authors contributed equally to this work.
¤ Current address: Department of Nursing and Health Sciences, Faculty of Health and Social Sciences, University of South-Eastern Norway, Porsgrunn, Norway
‡ These authors also contributed equally to this work.
* marit.muller@usn.no

**Data Availability Statement:** We consider our underlying minimal data set to contain sensitive

## Abstract

### Objective

To investigate whether physician-diagnosed asthma modifies the associations between multiple lifestyle factors, sick leave and work ability in a general working population.

### Methods

A cross-sectional study was conducted in Telemark County, Norway, in 2013. A sample of 16 099 respondents completed a self-administered questionnaire. We obtained complete data on lifestyle, work ability and sick leave for 10 355 employed persons aged 18–50 years. We modelled sick leave and work ability using multiple logistic regression, and introduced interaction terms to investigate whether associations with lifestyle factors were modified by asthma status.

### Results

Several lifestyle risk factors and a multiple lifestyle risk index were associated with sick leave and reduced work ability score among persons both with and without physician-diagnosed asthma. A stronger association between lifestyle and sick leave among persons with asthma was confirmed by including interaction terms in the analysis: moderate lifestyle risk score * asthma OR = 1.4 (95% CI 1.02–2.1); high lifestyle risk score * asthma OR = 1.6 (95% CI 1.1–2.3); very high lifestyle risk score * asthma OR = 1.6 (95% CI 0.97–2.7); obesity * asthma OR = 1.5 (95% CI 1.02–2.1); past smoking * asthma OR = 1.4 (95% CI 1.01–1.9); and current smoking * asthma OR = 1.4 (95% CI 1.03–2.0).

data, and also potentially identifiable individuals. Sharing restrictions on the minimal data set are imposed from: The Regional Committee for Medical and Health Professional Research Ethics in South-east Norway (Study ID: REC 2012/1665), The Norwegian Data Inspectorate and the Telemark Hospital Department of Research and Development. However, data may be shared for researchers who meet the criteria for access to confidential data upon request to the head of the Telemark-Study steering committee: Dr. Trude K. Fossum, Department of Occupational and Environmental Medicine, Telemark Hospital, Post box 2900 Kjøbekk, 3710 Skien. E-mail: fotr@sthf. no The minimal data set identification: Minimal. Lifestyle.WAS.SL.asthma.sav.

**Funding:** The work was supported by the University of South-Eastern Norway and Telemark Hospital.

**Competing interests:** The authors have declared that no competing interests exist.

There was no significant difference in the association between lifestyle and work ability score among respondents with and without asthma.

## Conclusions

In the present study, we found that physician-diagnosed asthma modified the association between lifestyle risk factors and sick leave. Asthma status did not significantly modify these associations with reduced work ability score. The results indicate that lifestyle changes could be of particular importance for employees with asthma.

## Introduction

Asthma is a common respiratory disease, and one of the most common chronic diseases worldwide [1]. In Norway, asthma prevalence has increased markedly in the last 20 years [2], and was estimated at 11.5% in 2013 [3]. Although a large proportion of patients diagnosed with asthma are young [4], asthma is also common among persons of working age [5]. Potential consequences for employees and employers, the health care system and wider society include low work ability [6–8] and loss of working days [9–15].

A person's lifestyle is known to have significant impact on health and well-being [16]. Good health is essential for work participation and endurance until retirement [17]. Limited studies have been conducted of modifiable lifestyle factors which may reduce sick leave and increase work ability among persons with asthma. A recent Dutch study explored the association between several chronic diseases, including respiratory diseases, selected lifestyle factors, sick leave and work ability among health care workers [18]. The study suggests that smoking and obesity negatively influence work ability, especially among persons with respiratory disease [18]. Another European study suggests that low physical activity and smoking are associated with sick leave among persons with respiratory diseases [19]. However, these studies focused on respiratory diseases in general, and did not specifically assess asthma.

Lifestyle risk factors often occur simultaneously [20]. Previous studies of associations between lifestyle factors and sick leave/work ability have largely examined a limited number of lifestyle factors. We have previously reported independent and additive relationships between multiple lifestyle risk factors (obesity, smoking, unhealthy diet and low physical activity) and low work ability in a large sample taken from the general working population in Norway [21]. However, few general population studies have examined multiple lifestyle risk factors and absence from work, and we are unaware of any studies which link these associations with asthma status.

The aim of this study was therefore to investigate whether physician-diagnosed asthma modifies associations of multiple lifestyle factors with sick leave and work ability in a general working population.

## Material and methods

### Study sample and population

The Telemark Study is a longitudinal population-based study conducted in south-eastern Norway. The initial cross-sectional part of the study was carried out in 2013, and consisted of a postal questionnaire mailed to a random sample (18–50 years of age) of the general population. The total population is approximately 170 000. Out of 48 142 eligible participants, 16 099

responded to the questionnaire. The response rate of 33% has occasioned a study on non-response [22].

For the present study, participants (18–50 years of age) were included if they had been employed in the past 12 months and had provided complete data on lifestyle risk factors (diet, physical activity, body mass index and smoking habits), and reported information about sick leave and work ability score. Complete data were available for 10 355 participants. Of these, 1 110 (11%) reported having physician-diagnosed asthma (Fig 1).

## Dependent variables

**Sick leave.** Sick leave was defined as one or more days on sick leave in the previous 12 months, confirmed by an affirmative answer to the question: "Have you been on sick leave over the course of the last 12 months?"

**Work ability score (WAS).** The work ability score derives from the work ability index developed by the Finnish Institute of Occupational Health [23]. We decided to use the first-item question of the work ability index, the work ability score (WAS): "Current work ability compared with the lifetime best", where a score of 0 represents complete work disability and a score of 10 represents work ability at its best. The WAS has been validated by previous studies [24, 25]. We dichotomised the variable into poor (0–7) and good (8–10) WAS.

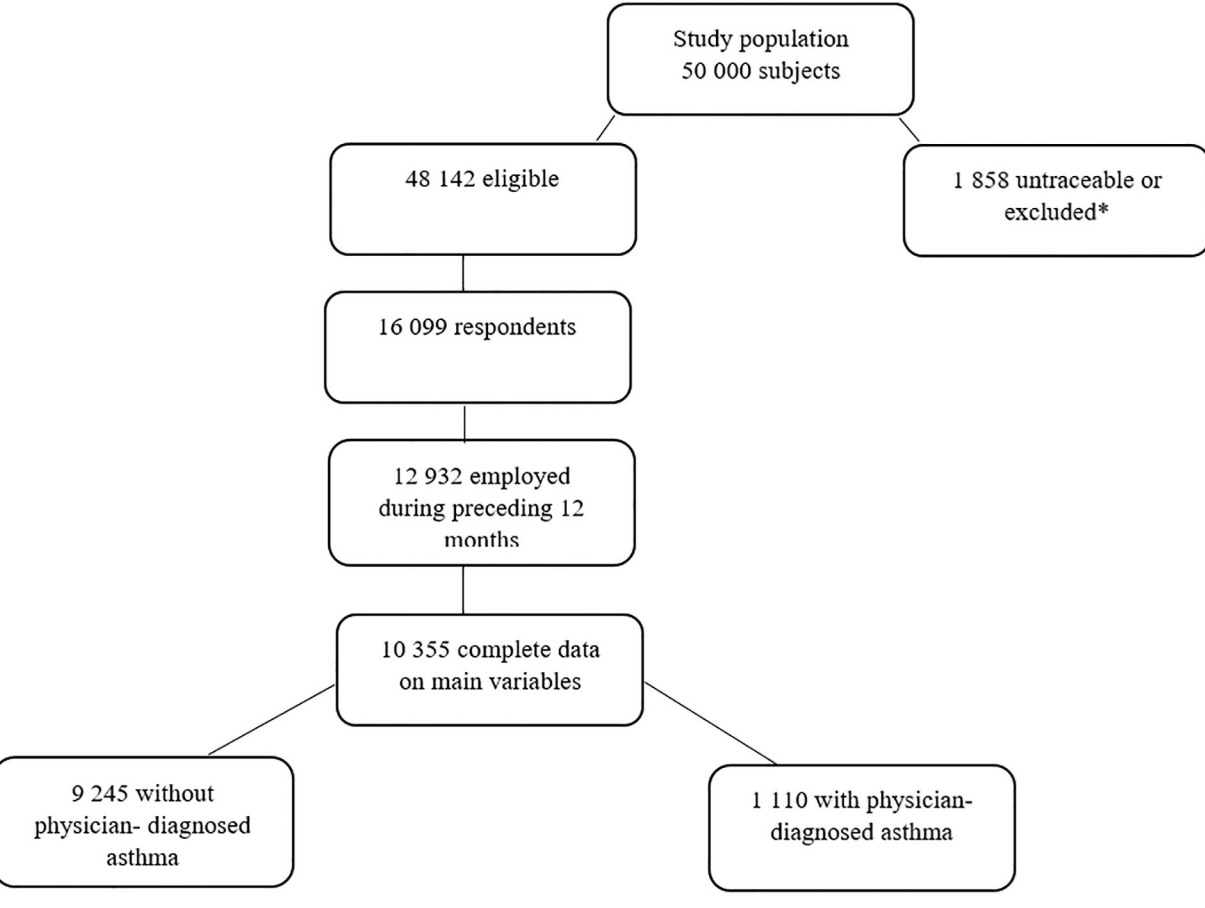

**Fig 1. Unknown address or language problems.**

## Independent variables

**Physician-diagnosed asthma.**   Participants were defined as having asthma if they responded affirmatively to the question: "Has a physician ever diagnosed you with asthma?"

## Lifestyle risk factors

**Diet.**   Diet was defined using food frequency questions previously validated and used in the Norwegian population-based Nord-Trøndelag Health Study (HUNT3) (2006–2008) [26, 27]. To reflect general dietary advice for improved health [28], a dietary sum score was constructed based on intake of fruits/berries and vegetables, fatty fish, sausages/hamburgers and chocolate/candies [21]. The sum score for each participant (scale 0–4) was calculated by summing their scores for the four indicators, reflecting the number of recommendations met [29]. The diet score was trichotomised into the categories low (0–1), moderate (2) and high (3–4) adherence to the general dietary recommendations. The three categories were labelled "unhealthy diet", "average diet" and "healthy diet", respectively.

**Physical activity.**   Moderate to vigorous leisure-time physical activity (MVPA) was assessed using validated questions and cut-off points covering frequency, intensity and duration of exercise as used in the HUNT1 (1984–1986) and HUNT3 (2006–2008) studies [30]. To reflect recommendations on adult MVPA ($\geq$ 150 minutes/week) [28], the responses to the questions regarding frequency, intensity and duration were combined to give a total MVPA score [30]. This was labelled "physical activity" and dichotomised into "active" and "less active". The weighted scores used to calculate the total score and the cut-off point emulating recommended MVPA were set according to the values used in the HUNT1 and HUNT3 studies [30, 31].

**Body mass index.**   Body mass index (BMI) was measured in accordance with the World Health Organization's cut-offs for different weight groups [32]. These were labelled "underweight" ($< 18.5$ kg/m$^2$), "normal weight" (18.5–24.9 kg/m$^2$), "overweight" (25–29.9 kg/m$^2$) and "obesity" ($\geq$30 kg/m$^2$).

**Smoking.**   Smoking was assessed by three questions. The first was: "Do you smoke every day?" Two follow-up questions were then asked: "Do you smoke occasionally?" and "If not, have you smoked in the past?" Smoking habits were divided into three categories labelled "current smoker" (every day and occasional smoking combined), "former smoker" and "never smoked".

**Lifestyle risk index.**   An overall lifestyle risk index was calculated to investigate the possible additive effect of lifestyle risk factors on work ability [21]. To estimate relative health risk, the individual lifestyle factors were given weighted risk scores: 0 (low health risk), 0.5 (intermediate health risk) and 1 (high health risk), and then summed into an overall index ranging from 0 to 4 (Table 1). To study different levels of lifestyle risk, the lifestyle risk index was divided into four categories: "low risk score" (total score 0–0.5), "moderate risk score" (total score 1–1.5), "high risk score" (total score 2–2.5) and "very high risk score" (total score 3–4). The index was labelled "Lifestyle risk index".

## Background variables

**Age.**   Age was categorised as "18–30", "31–40" and "41–50" years of age.

**Educational level.**   Educational level was categorised into three subgroups: "primary school + 1–2 years", "upper secondary and certificate" and "university/university college".

**Other chronic lung diseases.**   Participants were defined as having a chronic lung disease if they responded affirmatively to at least one of the following questions: "Has a physician ever

**Table 1. Study population characteristics, distribution of main variables and risk scores.**

| | Total | Without asthma | Asthma | |
|---|---|---|---|---|
| | n = 10 355 (100%) | n = 9 245 (100%) | n = 1 110 (100%) | |
| **Gender** | | | | |
| Male | 4 774 (46) | 4 276 (46) | 498 (45) | |
| Female | 5 581 (54) | 4 969 (54) | 612 (55) | |
| **Age groups** | | | | |
| 18–30 | 2 708 (26) | 2 378 (26) | 330 (30) | |
| 31–40 | 2 964 (29) | 2 647 (29) | 317 (29) | |
| 41–50 | 4 683 (45) | 4 220 (45) | 463 (41) | |
| **Educational level** | | | | |
| Primary school and lower secondary education (10 years or less) | 1 018 (10) | 923 (10) | 95 (9) | |
| Upper secondary education (an additional three to four years) | 4 242 (41) | 3 781 (41) | 461 (41) | |
| University or university college | 4 794 (46) | 4 272 (46) | 522 (47) | |
| Missing | 301 (3) | 269 (3) | 32 (3) | |
| **Other chronic lung diseases** | | | | |
| Yes | 223 (2) | 112 (1) | 111 (10) | |
| No | 10 132 (98) | 9 133 (99) | 999 (90) | |
| **Sick leave** | | | | |
| No sick leave in the previous 12 months | 7 023 (68) | 6 365 (69) | 658 (59) | |
| Sick leave in the previous 12 months | 3 332 (32) | 2 880 (31) | 452 (41) | |
| **WAS** | | | | |
| Good WAS (8–10) | 8 976 (87) | 8 064 (87) | 912 (82) | |
| Low WAS (0–7) | 1 379 (13) | 1 181 (13) | 198 (18) | |
| **Lifestyle risk factors** | | | | Lifestyle index risk score* |
| **Diet** | | | | |
| Healthy | 5 851 (56) | 5 246 (57) | 605 (55) | (0) |
| Average | 3 700 (36) | 3 287 (36) | 413 (37) | (0.5) |
| Unhealthy | 804 (8) | 712 (7) | 92 (8) | (1) |
| **Physical activity** | | | | |
| Active | 5 332 (51) | 4 732 (51) | 600 (54) | (0) |
| Less active | 5 023 (49) | 4 513 (49) | 510 (46) | (1) |
| **BMI category** | | | | |
| Normal weight (18.5–24.9 kg/m$^2$) | 4 951 (48) | 4481 (49) | 470 (42) | (0) |
| Underweight (<18.5 kg/m$^2$) | 128 (1) | 113 (1) | 15 (1) | (0.5) |
| Overweight (25–30 kg/m$^2$) | 3 733 (36) | 3 327 (36) | 406 (37) | (0.5) |
| Obese (>30 kg/m$^2$) | 1 543 (15) | 1 324 (14) | 219 (20) | (1) |
| **Smoking status** | | | | |
| Never smoked | 5 555 (54) | 4 973 (54) | 582 (52) | (0) |
| Former smoker | 2 298 (22) | 2 033 (22) | 265 (24) | (0.5) |
| Current smoker | 2 502 (24) | 2 239 (24) | 263 (24) | (1) |
| **Lifestyle risk score** | | | | |
| Low risk (0–0.5) | 2 592 (25) | 2 322 (25) | 270 (24) | |
| Moderate risk (1–1.5) | 4 030 (39) | 3 600 (39) | 430 (39) | |
| High risk (2–2.5) | 2 895 (28) | 2 592 (28) | 303 (27) | |
| Very high risk (3–4) | 838 (8) | 731 (8) | 107 (10) | |

* The numbers in brackets are the risk scores used for each variable when calculating the lifestyle risk index.

diagnosed you with chronic obstructive lung disease?" or "Has a physician ever diagnosed you with any chronic lung disease other than chronic obstructive lung disease or asthma?"

## Statistical analyses

The phi coefficient was used to assess the correlation between sick leave and low work ability, while Spearman's rho was used to assess the correlation between the individual lifestyle risk factors. We used a multiple logistic regression analysis to explore the association between individual lifestyle risk factors (diet, physical activity, body mass index and smoking), the multiple-lifestyle index (independent variables), and sick leave and work ability (dependent variables). Odds ratios (OR) with 95% confidence intervals were calculated for the likelihood of sick leave and low work ability. The individual lifestyle risk factors were adjusted for each other in the respective models. Age, gender, educational level and other chronic lung diseases were included as adjustment variables. In addition, sensitivity analyses were performed to adjust for other chronic diseases (cardiovascular disease, diabetes and mental illness).

To investigate whether asthma was a potential effect modifier, we decided to assess associations between independent and multiple lifestyle risk factors and sick leave and work ability stratified by asthma status.

It might be hypothesized that the effect of the combination of multiple lifestyle risk factors with asthma is greater than the sum of their separate effects (Fig 2). For further exploration of the data, interaction terms were included in logistic regression models (asthma multiplied

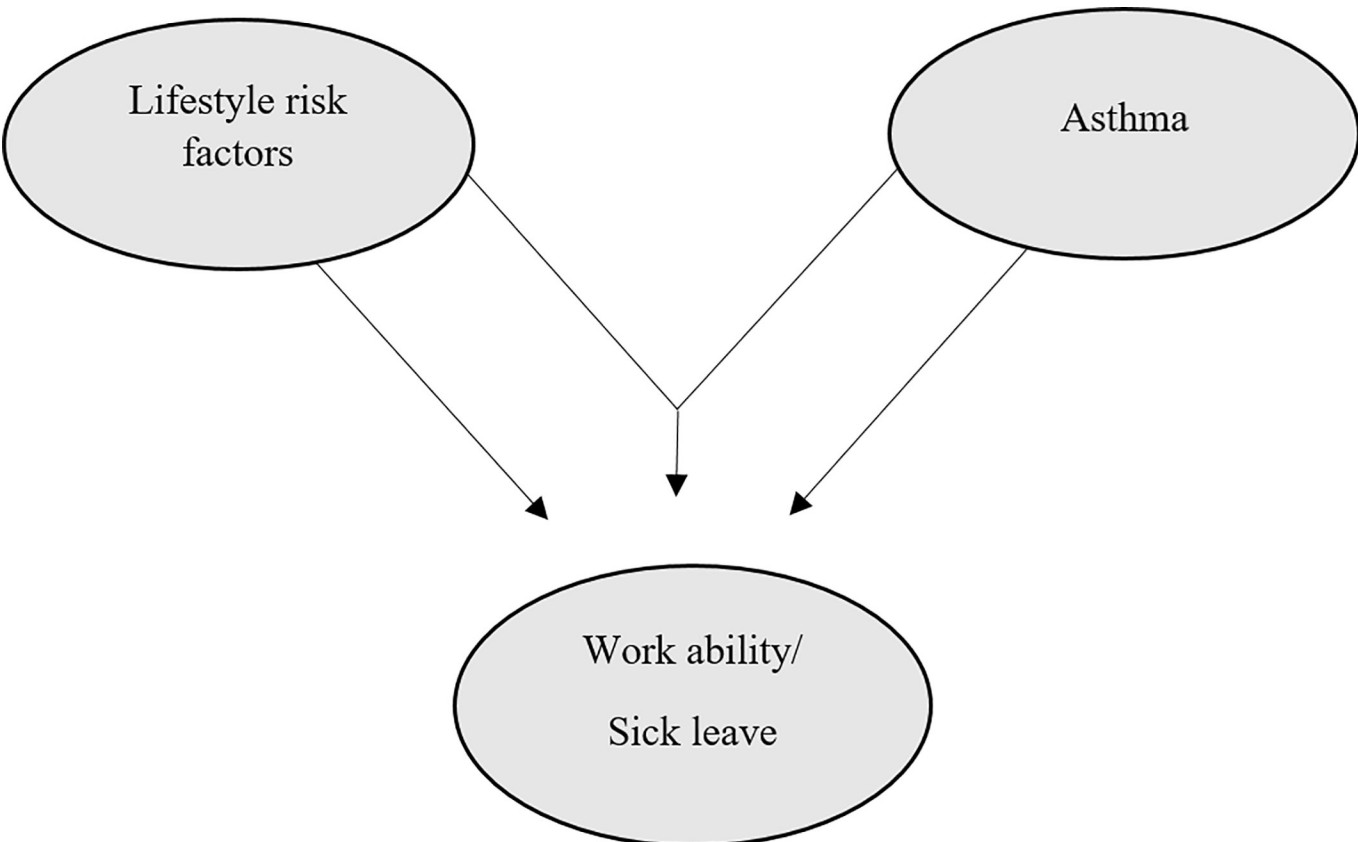

**Fig 2. Hypothesized interaction of lifestyle risk factors and asthma on sick leave/work ability.** Figure is inspired by Tonnon et al [33].

with each lifestyle risk factor and the lifestyle risk index, respectively) to reveal any multiplicative interaction.

The statistical analyses were carried out using IBM SPSS Statistics for Windows, version 24–2017. In all analyses, statistical significance was defined as $p < 0.05$ or a 95% confidence interval that did not include the null value.

## Ethical considerations

Written informed consent was obtained from all the participants. The study was conducted in accordance with the guidelines laid down in the Declaration of Helsinki, and were approved by the Regional Committee for Ethics in Medical Research (REC 2012/1665) and the Norwegian Data Protection Authority.

## Patient and public involvement

This study consulted user representatives from the Norwegian Asthma and Allergy Association in relation to study planning, study design and knowledge transfer. User representatives were included in the communication of results to health care workers, policy makers and the public on various media platforms (newspapers, internet, radio and television). One user representative served as a member of the study steering committee, and made important contributions to questionnaire development.

## Results

The study population characteristics are presented in Table 1. The sample consisted of 10 355 persons from the general working population of Telemark County (aged 18–50; slightly more female than male participants (54% vs 46%)). One-third of the subjects reported sick leave days in the past 12 months (32%). Most participants reported a good work ability (68%). Finally, 11% reported physician-diagnosed asthma.

Sick leave and work ability were weakly correlated (phi correlation 0.20). Spearman's rho correlations between individual lifestyle-related risk factors ranged from 0.03 between BMI and diet to 0.12 between low physical activity and diet.

Multiple logistic regression analysis showed that lifestyle risk score, BMI category and smoking status were all significantly associated with sick leave in the past 12 months, among persons both with and without asthma (Table 2). All associations were observed to be stronger among persons with asthma than persons without asthma. A stronger positive association between lifestyle and sick leave among persons with asthma was confirmed by including interaction terms in the analysis: (Lifestyle risk score (moderate) * asthma OR = 1.4 (95% CI 1.02–2.1); Lifestyle risk score (high) * asthma OR = 1.6 (95% CI 1.1–2.3); Lifestyle risk score (very high) * asthma OR = 1.6 (95% CI 0.97–2.7); Obesity * asthma OR = 1.5 (95% CI 1.02–2.1); Past smoking * asthma OR = 1.4 (95% CI 1.01–1.9); and Current smoking * asthma OR = 1.4 (95% CI 1.03–2.0).

Lifestyle risk score and physical activity were significant associated with WAS among persons both with and without asthma (Table 3). The associations were observed to be somewhat stronger among persons with asthma than among persons without asthma. A model including interaction terms between lifestyle and asthma showed that the trend of stronger association between lifestyle and WAS among persons with asthma, compared to persons without asthma, was not statistically significant.

**Table 2. Associations between lifestyle factors and sick leave by asthma status (n = 10 355).**

|  | Without asthma | Asthma |
|---|---|---|
| **Lifestyle risk index** | OR[a] (95% C.I.) | OR[a] (95% C.I.) |
| Low risk score (0–0.5) | 1.0 | 1.0 |
| Moderate risk score (1–1.5) | **1.2 (1.1, 1.4)** | **1.7 (1.2, 2.4)** |
| High risk score (2–2.5) | **1.4 (1.2, 1.6)** | **2.1 (1.4, 3.0)** |
| Very high risk score (3–4) | **1.8 (1.5, 2.1)** | **2.6 (1.6, 4.2)** |
| **Lifestyle risk factor** | OR[a] (95% C.I.) | OR[a] (95% C.I.) |
| Diet |  |  |
| Healthy | 1.0 | 1.0 |
| Average | 1.0 (0.92, 1.1) | 1.0 (0.79, 1.4) |
| Unhealthy | 1.1 (0.92, 1.3) | 0.93 (0.57, 1.5) |
| Physical activity |  |  |
| Active | 1.0 | 1.0 |
| Less active | 1.1 (0.97, 1.2) | 1.1 (0.85, 1.4) |
| Body mass index |  |  |
| Normal weight | 1.0 | 1.0 |
| Underweight ($<18.5$ kg/m$^2$) | 1.1 (0.76, 1.7) | 1.8 (0.60, 5.2) |
| Overweight (25–30 kg/m$^2$) | **1.2 (1.1, 1.3)** | 1.2 (0.92, 1.7) |
| Obese ($>30$ kg/m$^2$) | **1.6 (1.4, 1.8)** | **2.2 (1.5, 3.1)** |
| Smoker |  |  |
| Never smoked | 1.0 | 1.0 |
| Former smoker | **1.3 (1.2, 1.5)** | **1.7 (1.3, 2.4)** |
| Current smoker | **1.3 (1.2, 1.5)** | **1.7 (1.3, 2.4)** |

[a] Adjusted for gender, age, educational level and other chronic lung diseases.

## Discussion

In the present study, multiple lifestyle risk factors were associated with sick leave and reduced work ability among persons both with and without physician-diagnosed asthma. Most importantly, the association between multiple lifestyle risk factors and sick leave was modified by physician-diagnosed asthma. For persons with asthma, the lifestyle risk factors obesity, former smoker and current smoking were associated with sick leave, while low physical activity was associated with low WAS.

A direct comparison with other studies is challenging due to differences in study design and study populations, as well as this study's focus on asthma. Nonetheless, some similarities and differences should be acknowledged.

In the present study, we observed an association between increasing lifestyle risk scores and sick leave, especially among persons with asthma. To the best of our knowledge, no previous study has assessed the association between a multiple lifestyle risk index and sick leave in a general working population. However, two recent European studies [18, 19] have assessed individual lifestyle risk factors and associations with sick leave among health care workers. A multi-cohort study found that lifestyle factors such as smoking and low physical activity were associated with sickness absence linked to respiratory disease [19], while a Dutch study did not find any significant associations between individual lifestyle factors and sick leave among persons with respiratory diseases [18]. Unlike the present study, however, these studies did not specify which respiratory diseases were under investigation, and did not assess the strength of possible interactions with lifestyle factors. This makes comparison challenging.

**Table 3. Associations between lifestyle factors and work ability score by asthma status (n = 10 355).**

| | Without asthma | Asthma |
|---|---|---|
| **Lifestyle risk index** | OR[a] (95% C.I.) | OR[a] (95% C.I.) |
| Low risk score (0–0.5) | 1.0 | 1.0 |
| Moderate risk score (1–1.5) | **1.3 (1.1, 1.6)** | 1.5 (0.91, 2.4) |
| High risk score (2–2.5) | **1.9 (1.5, 2.2)** | **2.2 (1.3, 3.6)** |
| Very high risk score (3–4) | **2.3 (1.8, 3.0)** | **2.7 (1.5, 5.0)** |
| **Lifestyle risk factor** | OR[a] (95% C.I.) | OR[a] (95% C.I.) |
| Diet | | |
| Healthy | 1.0 | 1.0 |
| Average | 1.1 (0.99, 1.3) | 0.99 (0.71, 1.4) |
| Unhealthy | **1.3 (1.02, 1.6)** | 1.2 (0.65, 2.0) |
| Physical activity | | |
| Active | 1.0 | 1.0 |
| Less active | **1.4 (1.2, 1.6)** | **1.6 (1.2, 2.2)** |
| Body mass index | | |
| Normal weight | 1.0 | 1.0 |
| Underweight ($<18.5$ kg/m$^2$) | 1.3 (0.79, 2.2) | 1.6 (0.41, 5.9) |
| Overweight (25–30 kg/m$^2$) | 1.1 (0.94, 1.3) | 1.2 (0.81, 1.7) |
| Obese ($>30$ kg/m$^2$) | **1.4 (1.2, 1.7)** | 1.4 (0.89, 2.1) |
| Smoking | | |
| Never smoked | 1.0 | 1.0 |
| Former smoker | **1.2 (1.03, 1.4)** | 1.3 (0.85, 1.9) |
| Current smoker | **1.3 (1.1, 1.5)** | 1.4 (0.97, 2.1) |

[a] Adjusted for gender, age, educational level and other chronic lung diseases.

The observed modification by asthma status on the association between lifestyle risk score and sick leave was confirmed through the inclusion of interaction terms between lifestyle factors and asthma, suggesting the presence of multiplicative interactions. Our results indicate that persons with asthma could be more susceptible to sick leave due to lifestyle. This in turn suggests that lifestyle changes may be of particular importance to prevent sick leave among persons with asthma, even when few lifestyle risk factors are present.

As regards the individual lifestyle risk factors studied, our findings indicate that obesity is more strongly associated with sick leave among persons with asthma than among persons without asthma. This is consistent with previous literature [11, 15, 34]. Some studies have shown improvement in asthma outcomes following weight reduction, indicating potential benefits for the working life of these respondents [35, 36].

We also found that former and current smoking were more strongly associated with sick leave among subjects with asthma compared to persons without asthma. This is consistent with Swedish and Danish study results linked to current smoking [9, 15]. Interestingly, a Spanish cross-sectional study of persons with asthma [10] found that former smoking was associated with sick leave, but could not confirm an association between current smoking and sick leave [10]. The authors suggest the "healthy smoker effect" as a possible explanation for the results, implying that persons who smoke and have few respiratory symptoms continue smoking [10]. However, our study indicates that both past and current smoking may increase the likelihood of sickness absence, especially among persons with asthma.

Others have shown that factors such as age [6, 9, 13], occupation [8, 9], socio-economic status [14] and severity of asthma [10] are important predictive variables with regard to sick leave

and low work ability among persons with asthma. We therefore adjusted for age and education, but this did not significantly influence our results.

No significant modification by asthma status on the association between lifestyle and WAS was observed. However, statistically significant associations between lifestyle risk score and low WAS were observed among respondents both with and without asthma. Of the individual lifestyle factors, only low physical activity was significantly associated with low WAS among workers with asthma. Recent studies suggest that physical activity improves asthma control and lung function among adults with asthma [37, 38]. One possible explanation for our findings may be that low physical activity has an opposite, adverse effect and may therefore reduce self-perceived work ability. Moreover, a non-significant trend of decreased WAS was observed among smokers. Our results are consistent with a longitudinal Finish study of men diagnosed with asthma from youth, in which current smoking was associated with low work ability [7]. Moreover, a Dutch study [18] suggested a stronger association between smoking and low WAS among health care workers with respiratory diseases than among healthy individuals. However, as mentioned above, this study did not distinguish between different respiratory diseases or assess interactions [18]. Our findings provide additional insight into the association between multiple lifestyle factors and work ability among persons with asthma.

Our study indicates that persons with asthma may have greater benefits from lifestyle improvements than persons without asthma. According to our results, moderate lifestyle improvements could potentially decrease the likelihood of sick leave for this group of employees. Lifestyle is theoretically modifiable, and our findings imply that workplace measures targeting lifestyle changes may have a beneficial impact on persons both with and without asthma.

This study has strengths but also limitations that should be considered.

Important strengths are the inclusion of multiple lifestyle risk factors and the large study sample from the general working population. Other studies have focused on individual lifestyle risk factors in selected groups, such as subjects with asthma [10, 13], subjects in a specific occupation [18] and, often, male-only cohorts [7]. A further strength of this study is the use of validated questions which have also been used in other Norwegian cohort studies, for both independent (lifestyle risk factors) and dependent variables (sick leave).

Furthermore, several adjustment variables (age, gender, educational level and other chronic lung diseases) were included in the regression analyses. This adjustment did not alter the estimates substantially, indicating independent associations and limited risk of mistaken adjustment for intermediate variables in relevant causal pathways. Moreover, adjustment for other medical conditions (cardiovascular disease, diabetes and mental illness) did not significantly alter the results linked to sick leave and WAS.

The possibility cannot be excluded that we have underestimated the associations between lifestyle risk factors and sick leave and work ability due to the inclusion of all subjects with physician-diagnosed asthma, without differentiation based on severity or time of onset. Further, we did not analyse treatment parameters or medication use for persons with asthma, which may have influenced the associations. Studies have shown that persons on daily oral corticosteroids have less absenteeism than persons without such treatment [10].

Female, older age groups (41–50 years old) and more highly educated persons were slightly over-represented among the questionnaire respondents, indicating a possible selection bias. However, all regression analyses were adjusted for these variables. Generalisation of the results may be challenging due to the low response rate (33%). Nevertheless, analyses of non-responders indicate similar results to those of responders [22], and the associations appear less likely to be influenced by selection bias [39, 40].

One limitation of the study may be the use of self-reported physician-diagnosed asthma, which does not allow for verification of the diagnosis. Nevertheless, self-reported physician-

diagnosed asthma has been shown to have high specificity [41]. Lifestyle-related factors may be perceived as sensitive information. This could introduce a social desirability bias which obscures associations, for example due to underreporting of bodyweight [42]. The study design does not permit objective confirmation of respondent answers. However, the use of validated questions should reduce the likelihood of such bias.

Given the uncertainty about the temporal sequence of events that is inherent to the cross-sectional design, this study cannot claim causal relationships between lifestyle factors and sick leave or work ability.

## Conclusion

In the present study, we found that physician-diagnosed asthma modified the association between lifestyle risk factors and sick leave. Asthma status did not significantly modify these associations with reduced WAS. The results indicate that lifestyle changes may be particularly important for employees with asthma. These findings are significant for public health promotion and occupational intervention programmes aimed at preventing sick leave and improving work ability, especially among persons with asthma. Longitudinal studies should be conducted to explore these associations further.

## Supporting information

**S1 Data. Questionnaire Norwegian.**
(PDF)

**S2 Data. Questionnaire English.**
(DOCX)

**S1 Checklist. STROBE statement—Checklist of items that should be included in reports of *cross-sectional studies*.**
(DOCX)

## Acknowledgments

The authors wish to thank Regine Abrahamsen, Geir Klepaker and Gølin Finkenhagen Gundersen for assistance with data collection.

## Author Contributions

**Conceptualization:** Anne Kristin Møller Fell, Martin Veel Svendsen, Johny Kongerud.

**Formal analysis:** Marit Müller De Bortoli, Martin Veel Svendsen.

**Investigation:** Anne Kristin Møller Fell, Martin Veel Svendsen.

**Project administration:** Anne Kristin Møller Fell, Martin Veel Svendsen.

**Supervision:** Anne Kristin Møller Fell, Martin Veel Svendsen, Inger M. Oellingrath.

**Writing – original draft:** Marit Müller De Bortoli, Anne Kristin Møller Fell, Martin Veel Svendsen, Paul K. Henneberger, Johny Kongerud, Inger M. Oellingrath.

**Writing – review & editing:** Marit Müller De Bortoli.

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
