## [Decision Letter · Decision Letter 0]

19 Feb 2020

PONE-D-19-34978

Lifestyle, sick leave and work ability among Norwegian employees with asthma – a population-based cross-sectional survey conducted in Telemark County, Norway.

PLOS ONE

Dear Mrs. De Bortoli,

Thank you for submitting your manuscript to PLOS ONE. After careful consideration, we feel that it has merit but does not fully meet PLOS ONE’s publication criteria as it currently stands. Therefore, we invite you to submit a revised version of the manuscript that addresses the points raised during the review process.

We would appreciate receiving your revised manuscript by Apr 04 2020 11:59PM. To enhance the reproducibility of your results, we recommend that if applicable you deposit your laboratory protocols in protocols.io, where a protocol can be assigned its own identifier (DOI) such that it can be cited independently in the future. For instructions see: http://journals.plos.org/plosone/s/submission-guidelines#loc-laboratory-protocols

We look forward to receiving your revised manuscript.

Kind regards,

Davor Plavec

Academic Editor

PLOS ONE

Additional Editor Comments (if provided):

Please revise your manuscript according to the suggestion of the reviewer.

Journal Requirements:

2. Please provide additional details regarding participant consent. In the ethics statement in the Methods and online submission information, please ensure that you have specified (1) whether consent was informed and (2) what type you obtained (for instance, written or verbal). If your study included minors, state whether you obtained consent from parents or guardians. If the need for consent was waived by the ethics committee, please include this information.

Reviewers' comments:

Reviewer's Responses to Questions

**Comments to the Author**

1. Is the manuscript technically sound, and do the data support the conclusions?

Reviewer #1: Yes

2. Has the statistical analysis been performed appropriately and rigorously? 

Reviewer #1: Yes

3. Have the authors made all data underlying the findings in their manuscript fully available?

Reviewer #1: Yes

4. Is the manuscript presented in an intelligible fashion and written in standard English?

Reviewer #1: Yes

5. Review Comments to the Author

Reviewer #1: Thank you for the opportunity to review this paper. The manuscript is technically sound, the data support the conclusions and presented in an intelligible fashion and written in standard English. The paper gives a novelty insight how multiple lifestyle risk factors are associated to asthma and sick-leave. I would suggest the following – in order to make these interactions more understandable to your audience, I would suggest you make a diagram to illustrate these interactions. Also, I would suggest to always use the same wording throughout the whole manuscript- multiple lifestyle risk factors.

6. PLOS authors have the option to publish the peer review history of their article (what does this mean?). If published, this will include your full peer review and any attached files.

Reviewer #1: No

---

## [Author Response · Author response to Decision Letter 0]

25 Mar 2020

Reviewer #1: Thank you for the opportunity to review this paper. The manuscript is technically sound, the data support the conclusions and presented in an intelligible fashion and written in standard English. The paper gives a novelty insight how multiple lifestyle risk factors are associated to asthma and sick-leave. I would suggest the following – in order to make these interactions more understandable to your audience, I would suggest you make a diagram to illustrate these interactions. Also, I would suggest to always use the same wording throughout the whole manuscript- multiple lifestyle risk factors.

Answer: Thank you for these important comments. We have added a figure (Fig 2) in the manuscript illustrating the hypothesized interaction between lifestyle risk factors- asthma- sick leave/work ability. See page 8 and attached figure. We have obtained license from Springer Nature since the figure is inspired by an article from one of their Journals; Int Arch Occup Environ Health. 2019;92(6):855-64. License Number 4790660096986, License date March 16, 2020.

We have also corrected the wording to make the whole manuscript consistent with the wording of “multiple lifestyle risk factors”.

One behalf of the authors,

Marit Müller De Bortoli

---

## [Editor Report · Decision Letter 1]

31 Mar 2020

Lifestyle, sick leave and work ability among Norwegian employees with asthma – a population-based cross-sectional survey conducted in Telemark County, Norway.

PONE-D-19-34978R1

Dear Dr. De Bortoli,

We are pleased to inform you that your manuscript has been judged scientifically suitable for publication and will be formally accepted for publication once it complies with all outstanding technical requirements.

With kind regards,

Davor Plavec, MD, MSc, PhD, Prof.

Academic Editor

PLOS ONE

Additional Editor Comments (optional):

The manuscript is acceptable for publication in its current form.
---

## [Editor Report · Acceptance letter]

3 Apr 2020

PONE-D-19-34978R1 

Lifestyle, sick leave and work ability among Norwegian employees with asthma – a population-based cross-sectional survey conducted in Telemark County, Norway. 

Dear Dr. De Bortoli:

I am pleased to inform you that your manuscript has been deemed suitable for publication in PLOS ONE. Congratulations! Your manuscript is now with our production department. 

With kind regards,

on behalf of

Dr. Davor Plavec 

Academic Editor

PLOS ONE